# Carbonation Resistance and Pore Structure of Mixed-Fiber-Reinforced Concrete Containing Fine Aggregates of Iron Ore Tailings

**DOI:** 10.3390/ma15248992

**Published:** 2022-12-16

**Authors:** Wenbo Zheng, Sheliang Wang, Xiaoyi Quan, Yang Qu, Zhikai Mo, Changjun Lin

**Affiliations:** 1School of Civil Engineering, Xijing University, Xi’an 710123, China; 2School of Civil Engineering, Xi’an University of Architecture & Technology, Xi’an 710055, China

**Keywords:** hybrid fiber, iron tailings concrete, compressive strength, carbonation depth, pore structure

## Abstract

The disposal of industrial by-product tailings has become an important issue in solving environmental pollution. In this study, 15%, 30%, 50%, and 70% iron tailings were used to replace the natural sand in concrete, and 1.5% steel fiber and 0–0.75% PVA fibers were added to the iron tailings concrete. The effects of the iron tailings replacement rate and the fiber content on the mechanical properties, carbonization depth, and concrete porosity were studied in a carbonization environment. The results demonstrated that the compressive and splitting tensile strengths of concrete first increased and subsequently decreased with an increase in the iron tailings replacement rate, while the carbonation depth and porosity initially decreased and subsequently increased. When the replacement rate of iron tailings was 30%, the compressive strength and split tensile strength were increased by 7.6% and 17.7%, respectively, and the porosity was reduced by 8.9%. The compressive strength, carbonation depth and porosity of single-doped steel-fiber concrete were superior to those of ordinary iron tailings concrete. However, compared with single-doped steel fiber, the performance of steel-PVA fiber was further improved. Based on the mechanical properties, the carbonation depth test results of the three aforementioned types of concrete, the mathematical expression of the uniaxial compression stress–strain curve of iron tailings concrete, and the prediction equation of the carbonation depth of mixed-fiber iron tailings concrete were proposed. This study provides a reference for the application and popularization of fiber-reinforced iron tailings concrete in carbonization environments.

## 1. Introduction

With the rapid development of the steel industry and other industries, China produced 870 million tons of iron ore in 2020 (up 3.7% from the previous year), generating a large amount of iron tailings as a by-product. However, the utilization rate of iron tailings is very low, and only approximately 7% of them are recycled [1]. The produced iron tailings severely pollute the environment and occupy a relatively large land area. Various studies have shown [2,3,4,5] that iron tailings are increasingly used in the concrete industry as cement clinkers, coarse or fine aggregates, and mineral admixtures. Therefore, to protect the environment and ensure the effective waste utilization of iron tailings, they can potentially be applied in concrete preparation instead of natural sand.

Ail [6] replaced natural sand in concrete with iron tailings; the compressive strength, splitting tensile strength, and elastic modulus of the resulting material were higher than those of ordinary concrete. Francis [7] utilized iron tailings as fine aggregates for concrete with a low corrosion rate and acid erosion potential. Lv [8] proposed using iron tailings instead of natural sand in building materials. Owing to the high compressive strength, angular shape, and rough texture of iron tailings particles, the obtained iron tailings concrete exhibited better interfacial transition zone performance and durability than those of ordinary concrete. Cheng [9] used mechanochemically activated iron tailings to prepare concrete and studied its durability. The obtained results revealed that the anti-seepage and anti-freezing properties of the iron tailings concrete were superior to those of ordinary concrete at different replacement rates; however, its carbonation properties slightly deteriorated after adding iron tailings.

As an engineering material commonly used in building structures, ordinary concrete exhibits a high compressive strength but low tensile strength, ductility, and toughness, which are easily affected by the environment. To solve this problem, fiber-reinforced concrete has been widely examined. By adding fibers to the concrete matrix, it is possible to effectively control and delay the generation and expansion of cracks and increase its strength, ductility, and durability [10]. On the basis of the mechanical properties of the fiber, the fiber-reinforced concrete materials can be divided into high- and low-elastic-modulus fiber-reinforced concrete composites. According to the “mixing law” of composite materials, the elastic modulus of fiber-reinforced concrete increases with the increase in high-elastic-modulus fiber content, which is determined by the high elastic modulus of the fiber before cracking; large creep and deformation are easily produced in low-elastic-modulus fiber-reinforced concrete in the use stage, and deformation would be evident under the action of continuous high stress. At present, high- and low-elastic-modulus fiber-reinforced concrete composites are widely used and are representative of steel-fiber- and PVA-fiber-reinforced concrete composites [11]. Compared with single-doped fibers, hybrid fibers have multi-directional and multi-level features [12]. Yu [13] studied the mechanical properties of hybrid steel-PVA-fiber cement composites and concluded that their crack resistance, compressive strength, elastic modulus, and bending toughness exceeded those of the single-doped fiber concrete. Liu [14] discussed the influence of the mixed steel-PVA fiber content on the concrete tensile properties and found that the tensile strength of ordinary concrete increased by 37.5% at a steel fiber volume fraction of 1.5% and a polyvinyl alcohol (PVA) fiber content of 0.5%. Li [15] studied the effect of carbonation on the pore structure and splitting tensile strength of basalt–polypropylene fiber concrete; the obtained results demonstrated that the concrete pore structure changed after fiber addition, which increased its carbonation resistance.

Exposing concrete structures to a natural or harsh environment for a long time strongly affects their safety characteristics and usability; therefore, the durability of building structures is a very important parameter. Concrete carbonation is a common form of concrete deterioration. Although carbonation produces no adverse effects on concrete strength and may even increase its value, it ultimately causes the shrinkage and cracking of the concrete matrix, reduces concrete ductility, and leads to the destruction of the concrete structure and a reduction in its service life [16,17].

The main purpose of this paper was to study the effects of different iron tailings and fiber replacement rates on the mechanical properties and carbonization properties of concrete. The compressive strength, splitting tensile strength, carbonation depth, and porosity of iron tailings concrete and fiber-reinforced iron tailings concrete were evaluated. In order to protect the ecological environment and waste utilization, an appropriate amount of iron tail sand was used to replace natural sand to prepare iron tail sand concrete, and an appropriate amount of mixed fiber was added to enhance the mechanical properties and durability of the iron tail sand concrete in a carbonization environment to provide support for the engineering application of this material. The obtained results can help elucidate the carbonation mechanisms of iron tailings concrete and mixed-fiber concrete and provide a theoretical basis for the application of iron tailings and fibers in concrete preparation.

## 2. Materials and Methods

### 2.1. Materials

#### 2.1.1. Cementitious Materials

The cementitious materials used in this test were Qinling ordinary Portland cement and fly ash. The strength grade was PO42.5R. The chemical compositions and physical properties of the materials are presented in Table 1 and Table 2.

#### 2.1.2. Aggregates

The aggregates consisted of coarse aggregates and fine aggregates. The coarse aggregates were obtained from the crushed stone produced in the Xi ‘an area, with a particle size range of 5–25 mm (continuous gradation). The fine aggregates were composed of Chanhe natural sand and iron tailings, and their main components were analyzed by (Oberkochen, Germany) scanning electron microscopy and (Empyrean, The Netherlands) X-ray diffraction. Figure 1c shows the XRD pattern of the iron tailings. The main mineral composition of the iron tailings was quartz, with small amounts of hematite, anorthite, and muscovite. A detailed microscopic analysis and the physical properties of both aggregates are presented in Figure 1 and Table 3, respectively.

#### 2.1.3. Fibers

The fibers used in this study included steel and PVA fibers, whose appearance and physical properties are shown in Figure 2 and Table 4.

### 2.2. Mix Ratio

According to the replacement rates of the iron tailings and the fiber content when divided into ordinary concrete, iron tailings concrete, steel-fiber concrete, and steel–polyvinyl alcohol (PVA) fiber concrete, the substitution rates of iron tailings were 0%, 15%, 30%, 50%, and 70%. Research shows [18,19,20] that concrete performance is best when the steel fiber content is 1.5%, so the steel fiber content in this paper was 1.5%, and the PVA fiber contents were 0.25%, 0.5%, and 0.75%.

This paper refers to the specification requirements of the General Concrete Mix Design Code (JGJ55-2011) [21] after the preliminary trial and adjustment. The concrete mix design is shown in Table 5.

### 2.3. Preparation of Specimens

For ordinary concrete and iron tailings concrete, the specific preparation process was as follows: First, the quantitative coarse aggregate and fine aggregate were poured into the mixer and mixed dryly for 1 min. At this time, the coarse aggregate and fine aggregate were roughly evenly mixed. Second, the quantitative cement was slowly poured into the mixer and stirred for 1 min. Then, an appropriate amount of water-reducing agent was poured into a water container, evenly mixed with water, poured into a mixer, and stirred at a low speed for 2 min.

In the fiber-containing group of specimens, only when the fiber was uniformly dispersed in the concrete matrix could the bridging crack resistance of the fiber be fully exerted; the knotting and clumping of fibers were avoided as much as possible. Therefore, this test used a dry mixing after wet mixing method. The preparation process of the fiber-containing concrete consisted of the following steps: First, coarse aggregates, fine aggregates, and cement were poured into the mixer for 2 min of dry stirring. Afterwards, the manually rubbed and scattered fibers were slowly and evenly distributed in the resultant mix: first steel fibers (3 min of agitation) and then PVA fibers (2 min of mixing). Finally, water was mixed with the water-reducing agent and stirred for 2 min.

### 2.4. Analysis Procedure

#### 2.4.1. Mechanical Property Testing

The mechanical properties of the prepared concrete specimens were tested in accordance with the Standard Test Method for Mechanical Properties of Ordinary Concrete (GB/T50081-2019) [22] and the Standard Test Method for Fiber-reinforced Concrete (CESC13-2009) [23]. The cube compressive strength, splitting tensile strength, and axial compressive strength were determined by changing the collet on an MTS-microcomputer-controlled servo-hydraulic universal testing machine (E45.505, China).

#### 2.4.2. Carbonation Testing

Rapid carbonation was performed using the carbonization test chamber depicted in Figure 3a. Specimens were placed inside the chamber with a certain spacing to ensure the consistent carbonation conditions of their exposed surfaces. The carbon dioxide concentration was maintained at 20 ± 3%, the relative humidity was 70 ± 5%, and the temperature inside the box was maintained at 20 ± 2 °C. The carbonization depth was measured by a 1% phenolphthalein solution indicator. The phenolphthalein indicator reacted with the uncarbonated concrete to become red, and the reaction with the carbonized part of the concrete did not change color. The depth of the colorless part near the edge was measured as the carbonization depth. A calculation diagram of the carbonization depth is shown in Figure 3b.

#### 2.4.3. Nuclear Magnetic Resonance Studies

A MacroMR12-150H-I (Suzhou Niumai Analytical Instruments, Suzhou, China) nuclear magnetic resonance (NMR) instrument was used to determine the pore structure of the test block, including its T_2_ spectrum, porosity, and pore size distribution. NMR primarily monitors the signal produced by hydrogen atoms, which can be used for quantitative measurements of the water content. Before the test, the test block was immersed in water with a temperature of 20 ± 2 °C for 3 d to ensure the uniform distribution of water molecules in the concrete structure.

#### 2.4.4. SEM Observations

SEM was performed using a field-emission scanning electron microscope (Oberkochen, Germany), as shown in Figure 1. The principle of a scanning electron microscope is to produce physical signals such as secondary electrons by scanning the surface of a sample with a high-energy electron beam. It then uses secondary electron signal imaging to reflect the microstructure inside the concrete and the morphology of the erosion product on the surface of the specimen. First, the treated sample was placed on the conductive tape for gold spraying. Then, it was placed in the scanning electron microscope vacuum chamber to observe the imaging results.

## 3. Results and Discussion

### 3.1. Working Performance

The working performance of concrete is an important prerequisite for its excellent mechanical properties and durability [24]. The slump of concrete can quantitatively evaluate its working performance. The slump values of each mix ratio are shown in Figure 4.

Figure 4 shows that the slump of the NC group concrete was the highest. The slump of iron tailings concrete decreased with an increase in the iron tailings replacement rate, and the slump of T70 concrete was the lowest. The slump values decreased by 8.1%, 15.2%, 24%, and 36%, respectively, when the replacement ratios of iron tailings were 15%, 30%, 50%, and 70%. Similar changes have previously been reported [6,25]. This is mainly because iron tailings sand, compared with natural sand, has a larger specific surface area, a rougher particle texture, smaller fineness, and stronger water absorption [26]. The slump of fiber-reinforced concrete decreased with an increase in the PVA fiber content. Compared with ordinary concrete, when the contents of PVA fiber were 0%, 0.25%, 0.5%, and 0.75%, the slump of the concrete decreased by 31.5%, 40%, 46.3%, and 56.7%, respectively. This was mainly due to the fiber adhesion, tension, and water absorption [27], while the evenly dispersed fibers formed a spatial network structure with a viscous effect within the concrete, thereby inhibiting the flow of concrete mortar.

### 3.2. Cube Compressive Strength

As shown in Figure 5, when the replacement rate of iron tailings was equal to 30%, the concrete compressive strength reached a maximum, and its strength increased by 7.6% with respect to that of ordinary concrete. The compressive strengths of the T15, T50, and T70 concrete specimens increased by 4%, −5.8%, and −13.1%, respectively. The results by Sunil [28] showed that when the replacement rates of iron tailings were 30% and 50%, the strength increased by 9.19% and −5.43%, respectively, compared to ordinary concrete. Qu [29] reported that when the replacement rates of iron tailings were 40%, 80%, and 100%, the strength increased by 6.95%, 1.27%, and −24.56%, respectively, compared to ordinary concrete. Further, Ali [6] showed that when the replacement rates of iron tailings were 25%, 50%, 75%, and 100%, the strength increased by 12.9%, 10.5%, 10.2%, and 1.3%, respectively, compared to ordinary concrete.

There are several reasons for the increase in compressive strength after adding iron tailings: (1) The iron tailings particles are finer than those of natural sand. Appropriate iron tailings fill the pores inside the concrete, optimize the pore structure, and increase the concrete density. (2) The iron tailings material with a rough surface improves the bonding between the cement and aggregate interfaces, thus increasing the concrete strength. Meanwhile, the iron tailings particles with high iron concentrations also increase the concrete strength. However, with an increase in the replacement rate of iron tailings, the concrete strength decreases because iron tailings have a larger specific surface area than natural sand. In addition, the water demand of the concrete mixture increases, and its fluidity decreases, leading to the separation of the mortar and aggregates, which is not conducive to compact forming; as a result, the strength of the iron tailings concrete decreases [2,8].

As shown in Figure 6, fiber addition positively influenced the concrete compressive strength, and the effect of mixed steel–PVA fibers was stronger than that of single-doped steel fibers. The compressive strength of concrete increased first and then decreased with an increase in the PVA fiber content, and the concrete strength increased by 7.9%, 12%, 16%, and 10.6%, respectively. Zheng [30] reported that when the contents of steel fiber and PVA fiber were 1.25% and 0.75%, the strength of ordinary concrete increases by 17.4%.

The observed strength improvement in the fiber-reinforced concrete was mainly due to the increase in its crack resistance and toughening. The effect of introduced fibers on crack resistance mainly depends on their distribution in the concrete matrix. When fibers are evenly distributed in the matrix, they are strongly bonded to the slurry. When a crack expands, it can be quickly blocked by the adjacent steel fibers, preventing crack development and thereby increasing the crack resistance of the concrete specimen. The increased concrete toughness is due to the fiber’s bridging–debonding–sliding failure mode, which promotes energy dissipation. Moreover, the hydroxyl groups attached to the molecular chain of PVA fibers form strong bonds between the fibers and the concrete matrix. A large number of small PVA fibers increase their contact area with the concrete matrix, which facilitates stress transfer through cracks. In addition, they form a complex three-dimensional fiber network with a disordered distribution of steel fibers. Steel fibers inhibit the formation of large cracks in concrete, and PVA fibers inhibit early plastic shrinkage, dry shrinkage, and microcrack generation [15]. When the PVA fiber content exceeds a certain value, it becomes difficult for excessive PVA fibers to disperse, and an apparent clumping process occurs, resulting in the formation of new weak areas either without fibers or with low fiber amounts. The addition of fibers redistributes and transfers stress to the concrete matrix. If this stress distribution is not balanced, new stress concentration regions may be formed, leading to concrete cracking in the areas without fibers.

With an increase in the carbonation age, the concrete compressive strength stably grows. As shown in Figure 5 and Figure 6, during the first 7 d of carbonation, the concrete compressive strength exhibited the largest growth rate. The compressive strength of the iron tailings concrete specimens increased by 5.5%, 3.4%, 3.5%, 8.2%, and 9.8%, and the compressive strength of the fiber-reinforced concrete samples increased by 3.9%, 1.4%, 0.08%, and 3.9%. With an increase in the carbonation age, the growth of the compressive strength stabilized. Between 28 and 56 d, the strength of the iron tailings concrete increased by 1.7%, 1.7%, 1.2%, 2.0%, and 1.4%, and the compressive strength of the fiber concrete samples increased by 0.42%, 1.2%, 1.3%, and 1.9%.

After the carbonation of concrete, the chemical composition and pore structure of its matrix changes. CO_2_ reacts with Ca(OH)_2_ to generate water-insoluble calcium carbonate, which effectively fills the pores and microcracks inside the concrete matrix and reduces its porosity. With an increase in the carbonation age, the compactness of the concrete surface increases owing to the formation of dense calcite microcrystals on the surface of the carbonation layer, which inhibits the CO_2_ diffusion into the concrete matrix, decreasing the diffusion rate. Therefore, the concrete compressive strength first increases with an increase in the carbonation age and then gradually stabilizes. Note that carbonation has little influence on the compressive strength of the fiber-reinforced concrete because the addition of fibers enhances its internal structure and increases the concrete density, suppressing CO_2_ penetration.

### 3.3. Cube Splitting Tensile Strength

The splitting tensile strength of concrete first increased and then decreased with an increase in the iron tailings replacement rate, as shown in Figure 7. At an iron tailings replacement rate of 30%, the tensile strength reached a maximum, and the splitting tensile strengths of T15, T30, T50, and T70 increased by 14%, 17.7%, 12.6%, and −2.7%, respectively.

The tensile strength of concrete is not only an important index for determining its crack resistance but also a key parameter for indirectly measuring the bond strength between concrete components. In this work, the optimal bond strength was achieved at an iron tailings replacement rate of 30%. This concrete had the densest filling structure, resulting in improved bonding between the aggregate and the cement mortar. With an increase in the iron tailings replacement rate, the bonding strength between the aggregates decreased, leading to a decrease in the tensile strength.

As shown in Figure 8a, the addition of fibers significantly increased the concrete splitting tensile strength. The strengths of the concrete samples with fiber iron tailings were 15.9%, 19.1%, 22.9%, and 17.7% higher than that of the T30 iron tailings concrete.

The rough surface of iron tailings enhances the bonding of steel fibers; as result, the fibers are difficult to pull out or break. When the concrete is subjected to a large load, tensile stress is transferred to the steel fiber through the concrete due to a bridging effect between the steel fiber and concrete. During the concrete splitting tensile failure, steel fibers absorb a large amount of energy, which compensates for the deficiency of the concrete matrix tension. After adding PVA fibers, the tensile strength of the concrete further increases because of the lower fiber spacing in the concrete structure, which limits the development of primary cracks caused by segregation, shrinkage, and other processes. However, when the PVA fiber content exceeds 0.5%, the concrete tensile strength decreases. Similar to the trend observed for the compressive strength, the excessive addition of PVA fibers induces their agglomeration, leading to an uneven distribution of fibers in the concrete matrix. If the number of fibers at the splitting interface is too high or too low, stress concentration occurs, and the tensile strength is reduced.

The tensile strength of concrete always maintains a steady growth trend with increasing carbonation ages. As shown in Figure 8b, the strength of the iron tailings concrete increased by 7.8%, 6%, 4.6%, 6.4%, and 10.3% after 14 d of carbonation, while the strength of the fiber concrete increased by 6%, 4.1%, 6.1%, and 6.4% over the same period.

Based on the literature [31,32], the relationship between the compressive strength and the splitting tensile strength of concrete can be expressed by Equation (1):(1)$$f_{\mathrm{sts}}\;=\;af_{cu}^{b}$$
where a and b are the constants, and *f*_sts_ and *f*_cu_ are the splitting tensile and compressive strengths, respectively.

Based on the obtained experimental data, a statistical regression analysis was conducted to derive Equation (2):(2)$$f_{\mathrm{sts}}\;=\;0.23f_{cu}^{0.75}$$

Different national standards stipulate different calculation relationships between the compressive and splitting tensile strengths of concrete. According to the American Concrete Association standards ACI-363R and ACI-318-11 and the European Concrete Commission (CEB), the corresponding relationships are expressed by Equations (3)–(5), respectively:(3)$$f_{\mathrm{sts}}\;=\;0.59f_{cu}^{0.5}$$
(4)$$f_{\mathrm{sts}}\;=\;0.53f_{cu}^{0.5}$$
(5)$$f_{\mathrm{sts}}\;=\;0.3f_{cu}^{0.67}$$

As shown in Figure 9a, based on the test results of this study, these standards tend to overestimate the splitting tensile strength of iron tailings concrete. Equation (2) was obtained through a regression analysis, as shown in Figure 9b, and the error of the proposed formula was within the range of ±5%.

### 3.4. Axial Compressive Strength

Similar to the cube compressive strength and splitting tensile strength, the axial compressive strength of the prepared concrete specimens first increased and then decreased with an increase in the iron tailings replacement rate and PVA fiber content. According to Table 6, the strengths of the T15 and T30 iron tailings concrete samples increased by 7.3% and 14.5%, respectively; those of the T50 and T70 iron tailings concrete specimens decreased by 2.8% and 5.6%, respectively; and the strength of the fiber-reinforced concrete increased by 7.7%, 11%, 14.7%, and 9.3%. The axial compressive strength steadily increased with an increase in the carbonation age. After 28 d of carbonation, the strength of the iron tailings concrete increased by 14.4%, 16.1%, 9%, 16.3%, and 17.9%, and the strength of the fiber concrete increased by 13.8%, 12.1%, 10.1%, and 10.9%.

The elastic modulus represents the deformation resistance of the material under stress, that is, the stiffness of the material, while the compressive strength of concrete is achieved by homogenizing the constituent materials and their strength. An appropriate amount of iron tailings increases the compressive strength of concrete; as a result, the ability to resist elastic deformation is enhanced, and the elastic modulus increases.

The relationship between the axial compressive strength of concrete and its elastic modulus can be fitted by the least-squares method to obtain the coefficients a = 5521 and b = 0.5 (Figure 10 and Equation (5)). Different countries use different a, b, and c values. For example, in the ACI-363R standard, Ec = 3320fcu^0.5 + 6900, and the error between the experimental and calculated values in the ACI-318-11 standard is Ec = 4370fcu^0.5 (Table 7). The values calculated using the ACI-363R and ACI-318-11 standards are lower than the test values, and the corresponding errors are relatively large. In contrast, the errors between the experimental values and those calculated via Equation (5) in this study are small, ranging from −0.37% to 1.57%.
(6)$$Ec=5512f_{cu}^{0.5}\;\;\;\;\;\;\;R^{2}\;=\;0.92$$

The stress–strain curve of concrete under compression provides an experimental basis for structural design and nonlinear analysis. In this study, a constitutive relation of the iron tailings concrete was fitted by the standard stress–strain curve equation of ordinary concrete [33].

According to the GB50010–2010 Code for Design of Concrete Structures [34], the compressive stress–strain curve of ordinary concrete is a piecewise curve equation, and the following compressive stress–strain curve equation is proposed for the iron tailings concrete:y = ax + (3 − 2a)x^2^ + (a − 2)x^3^, 0 ≤ x ≤ 1(7)
y = x/(b(x − 1)^2^ + x), x ≥ 1(8)
where y = σ_c_/σ_p_, x = ε_c_/ε_p_, and σ_c_ and σ_p_ are the compressive stress and peak compressive stress, respectively. ε_c_ and ε_p_ are the compressive strain and peak compressive strain, respectively; a is the ascending stage parameter; and b is the descending stage parameter. The entire stress–strain curve of each group of concrete specimens was fitted with Equations (7) and (8), and the calculation formulas of a and b were obtained by the least-squares method:(9)$$a=33.638f_{cu}^{0.066}\;-\;40.595$$
(10)$$b=27104.09f_{cu}^{-2.965}$$

To verify the accuracy of the derived stress–strain curve equation and parameter calculation procedure, the fitting data were compared with the test results, as shown in Figure 11, which shows that the curve model proposed in this study can accurately predict the stress–strain curve of iron tailings concrete under axial compression.

### 3.5. Carbonation Depth

The carbonation depths of the studied concrete specimens are shown in Figure 12. According to Figure 13, this value increased with an increase in the carbonation age. Upon increasing the iron tailings replacement rate, the carbonation rate decreased first and then increased. When the iron tailings replacement rate reached 30%, the carbonation depth reached a minimum. After 7 d of carbonation, the carbonation depth of the iron tailings concrete decreased by 3.8%, 6%, −5.9%, and −6.1%, while after 28 d of carbonization, it decreased by 12.4%, 28.5%, 14.6%, and −12.5%.

Because the size of iron tailings particles is small, they can effectively fill the internal pores of concrete and make the concrete more compact; therefore, they increase the concrete carbonation resistance. In addition, iron tailings may potentially exhibit volcanic ash activity. With an increase in the carbonation age, the secondary hydration reaction between iron tailings and Ca(OH)_2_, the main hydration product of cement, results in the formation of calcium silicate hydrate and other compounds, thus improving the internal pore structure and concrete density. Furthermore, the water absorption of iron tailings is higher than that of natural sand; hence, iron tailings reduce the relative humidity inside the concrete, inhibiting the CO_2_ reaction and decreasing the carbonation depth [9,35,36].

The addition of fibers significantly reduced the concrete carbonation depth, as shown in Figure 12. After 28 d of carbonation, the carbonation depths of the fiber-reinforced concrete specimens decreased by 39.6%, 43.7%, 54.5%, and 41.3%.

The carbonation resistance of the fiber-reinforced concrete is mainly affected by the filling of internal pores and a reduction in the number of cracks. After a certain amount of steel fiber is added to the concrete, a complex three-dimensional disordered support system is formed in the matrix, which inhibits aggregate sinking, makes the mortar distribution more uniform, improves the concrete compactness, and enhances the concrete carbonation performance. However, as CO_2_ continues to diffuse, it may lead to the corrosion of steel fibers; the produced rust destroys the bonding between the fibers and mortar, promoting the infiltration and diffusion of CO_2_. The incorporation of PVA fibers into the concrete matrix further increases its carbonation resistance. Owing to the large density and coarse diameter of steel fibers, their volume fraction in the concrete structure is very large, which weakens the grid effect of these fibers and facilitates CO_2_ penetration. After adding PVA fibers, their volume decreases. However, PVA fibers are hydrophilic, which promotes the accumulation of a large amount of water on their surfaces, creates favorable conditions for cement hydration, inhibits the early formation of plastic shrinkage and dry shrinkage cracks, reduces the cracking temperature, and increases the crack resistance of the concrete [37,38].

### 3.6. Derivation of A Carbonation Model

The steel–PVA mixed-fiber iron tailings concrete is a new concrete-based composite material, and its carbonation proceeds similarly to the carbonation of ordinary concrete. In this study, the carbonation depth model of ordinary concrete was modified by considering the iron tailings replacement rate and the volume fractions of steel and PVA fibers. The empirical model assumes that carbonation depth is proportional to the square root of the carbonization time. Smolczyk established a calculation formula for the concrete carbonization depth and compressive strength based on experimental data and assumed that the carbonation depth was proportional to the reciprocal of the square root of the compressive strength. Based on these results, carbonation depth models were derived for the iron tailings concrete and steel–PVA fiber iron tailings (Equations (11) and (12), respectively) [39,40]:(11)$$X=\beta (\rho 1)\sqrt{\frac{c}{f_{u}{}_{1}}t}$$
(12)$$X=\beta (\rho 2)\sqrt{\frac{c}{f_{u}{}_{2}}t}$$
where X: concrete carbonation depth (mm); *f_u_*_1_: compressive strength of the iron tailings concrete before carbonation, MPa;*f_u_*_2_: compressive strength of the steel–PVA fiber iron tailings concrete before carbonation, MPa;*t*: carbonation time;*C*: CO_2_ concentration;β(ρ1): influence coefficient of iron tailings, which is a function of their replacement rate (ρ1);β(ρ2): influence coefficient of steel–PVA fiber iron tailings, which is a function of the PVA fiber content (ρ2).


Using the laws of mathematical statistics, the carbonation depth was expressed as a function of carbonation time, compressive strength, and CO_2_ concentration and subjected to an independent variable regression analysis, as shown in Figure 14. The carbonation depth fitting curves of different iron tailings replacement rates and fiber contents were calculated, and the regression analysis values of the influence coefficient (β) of different iron tailings replacement rates and fiber contents were obtained, as shown in Table 8. The carbonation influence coefficient is a nonlinear fitting parameter that can be expressed by the following equation:
β1 = (0.00569ρ12 − 0.33128ρ1 + 25.8499)(13)
β2 = (0.00124ρ22 − 0.10858ρ2 + 12.7005)(14)

Here, Equations (13) and (14) are substituted into Equations (11) and (12), which are the carbonation depth equations of the iron tailings concrete and steel–PVA fiber iron tailings concrete, respectively:(15)$$X=(0.00569\rho 12-0.33128\rho 1+25.8499)\sqrt{\frac{c}{f_{u}{}_{1}}t}\;\;\;\;\;\;\;R^{2}\;=\;0.97$$
(16)$$X=(0.00124\rho 22-0.10858\rho 2+12.7005)\sqrt{\frac{c}{f_{u}{}_{2}}t}\;\;\;\;\;\;\;R^{2}\;=\;0.96$$

By utilizing Equations (15) and (16), the carbonation depths of different iron tailings concrete and steel–PVA hybrid fiber concrete samples were calculated. The ratios of the calculated and experimental carbonation depth values are listed in Table 9.

The average value of μ = 0.96 and mean square deviation of σ = 0.099 were obtained for the ratio of the calculated carbonation depth of the iron tailings concrete to the experimental value. The average value of μ = 1.02 and mean square deviation of σ = 0.13 were obtained for the ratio of the calculated carbonation depth of the steel-PVA fiber-reinforced concrete to the experimental value, which indicate a good agreement of the developed carbonation model with the experimental data.

### 3.7. NMR Analysis

#### 3.7.1. Before Carbonation

The relaxation time of a T_2_ spectrum is related to the pore size, and the signal intensity reflects the number of pores. The T_2_ spectra of the concrete specimens fabricated at different replacement rates of iron tailings are shown in Figure 15. Each T_2_ spectrum is composed of one main and two small peaks. Compared with the control group (NC), the T_2_ spectrum of the T30 concrete was shifted to the left, and the relaxation time corresponding to the peak value and signal amplitude were decreased. Meanwhile, the T_2_ spectra of the T50 and T70 concrete specimens were shifted to the right, and their relaxation times corresponding to the peak values and signal amplitude were greater than those of the control group. These results confirm that an appropriate iron tailings replacement rate can reduce the number of concrete pores and increase the concrete density.

A change in the T_2_ spectral integral area reflects a change in the concrete pore volume. Table 10 lists the T_2_ spectral areas and relative fractions of the characteristic peaks obtained at different iron tailings substitution rates. According to Table 6, the spectral areas of the T15 and T30 groups decreased by 6.4% and 8.9%, while the spectral areas of the T50 and T70 groups increased by 14.9% and 32.1%. Meanwhile, with an increase in the iron tailings replacement rate, the area of the first peak initially increased and then decreased, whereas the areas of the second and third peaks exhibited the opposite trend. Moreover, the area of the first peak had the largest fraction, indicating that the small and medium pores in the concrete samples constituted a majority. Thus, the addition of an appropriate amount of iron tailings could considerably improve the pore structure. The pores with smaller sizes were easily filled, while the pores with larger sizes did not expand significantly, which improved the mechanical properties of the material. When the substitution rate exceeded 30%, the number of large pores in the concrete matrix increased owing to poor bonding between the aggregates.

Porosity is the most commonly used parameter for evaluating the pore characteristics of porous materials. Figure 16 depicts the porosity values of the concrete specimens obtained at four different iron tailings substitution rates. It shows that with an increase in the tailings content the porosity first decreased and then increased, which was consistent with the previous data. This trend indicates that an appropriate iron tailings content (not more than 30% in this study) helps reduce the concrete porosity. Because tailings particles are finer than natural sand and have a rough surface, they can effectively fill the pores inside the concrete structure and improve its compactness. This also confirms that iron tailings participate in the secondary hydration process and fill some small pores. However, when the content of iron tailings exceeds a critical value, excessive iron tailings weaken the microaggregation effect of the bonding system, decrease the hydration rate, and reduce the amount of hydration products, which becomes insufficient to fill the pores.

#### 3.7.2. After Carbonation

As shown in Figure 17, the T_2_ spectrum shifted to the left during the carbonation process, and the signal intensity continuously decreased. After seven days of carbonation, each peak of the T_2_ curve sharpened and decreased significantly. After 14 d of carbonation, the peaks continued the trend observed after 7 d of carbonation, and the amplitude of each peak value gradually decreased. When the carbonation age reached 56 d, both the second and third peaks were still detected (especially the third peak). However, the carbonation effect became stronger with an increase in the carbonation age, and the concrete pore structure was constantly optimized.

Table 11 lists the NMR T_2_ spectral areas and relative fractions of the characteristic peaks obtained at different carbonation ages. It shows that the area of the first peak in the T_2_ spectrum decreased and the fraction of the first peak in each group increased with increasing carbonization age. Furthermore, the areas and relative fractions of the second and third peaks decreased with increasing carbonization age. This indicates that the hydration products obtained during carbonation convert the large pores into small pores in the concrete structure and improve its properties.

Concrete porosity typically reflects the carbonation process. As shown in Figure 18, with an increase in the carbonation age, the porosity values of the concrete specimens decreased by 7.6%, 10.9%, 13.4%, and 15.1%. Furthermore, carbonation generates calcium carbonate, which makes the concrete sample more compact and fills its internal pores.

Compressive strength is the most basic mechanical property index of concrete. The fitting equation and regression parameters of the porosity–compressive strength model listed in Table 12 were obtained by performing a regression analysis of the NMR spectra of the concrete specimens with different iron tailings replacement rates, as shown in Figure 19.

Here, σ_c_ is the compressive strength and P is the total porosity of concrete.

To study the effect of the pore size on the concrete compressive strength, the pores were divided into gel pores (<10 nm), mesopores (10–100 nm), capillary pores, and macropores (>100 nm) [32,41]. Based on the experimental data obtained in this study, a statistical regression analysis was conducted to derive following equation:σ_c_ = 0.4618P < 10 − 0.169P10 − 100 − 0.4438P > 100 + 1(17)

The theoretical and experimental compressive strengths are compared in Figure 20. It shows that the calculated values were in good agreement with the experimental data.

### 3.8. SEM Observations

The failure zone of concrete under loading is usually located at the interface between the aggregate and cement paste, which is called an interfacial transition zone. This zone is a weak link in the concrete specimen, and cracks are preferentially initiated in the transition zone. As shown in Figure 21, the interfacial transition zone of the T30 iron tailings concrete was denser than that of ordinary concrete because the surface of the iron tailings particles was very rough, the bonding interface with the cementation system was serrated, and the bonding strength was high. The fineness of iron tailings is lower than that of natural sand; hence, they can be combined with coarse aggregates to form a more robust concrete skeleton. This result also indicates that the crack resistance of the iron tailings concrete is higher than that of ordinary concrete. With an increase in the iron tailings substitution rate, the number of cracks in the interfacial transition zone increased. This phenomenon occurred for the following reasons: On one hand, the water absorption rate of iron tailings is higher than that of natural sand; therefore, there is more water in the concrete matrix of iron tailings during the preparation process. This moisture evaporates from the matrix in the later stage to increase the number of cracks. On the other hand, owing to the small particle size and relatively large specific surface area of iron tailings, the bonding between the material and the concrete matrix is weakened, and the pores are increased [42]. 

According to the established fiber bonding mechanism, the matrix and fibers in the fiber-reinforced concrete interact with each other through chemical bonding at the interface to jointly bear the applied force. For this concrete type, the mechanical properties of the fiber–cement slurry interfacial transition zone represent a weak link affecting its macroscopic properties. When the concrete is subjected to an external force, shear stress is formed at the interface between the fibers and the matrix owing to the strain difference. This stress is continuously transmitted and weakened along the fiber direction until both the fibers and matrix deform synchronously. Therefore, the bond strength plays a decisive role in the reinforcement and crack resistance of fibers, and the fiber–matrix bond performance mainly depends on the bonding between the fibers and the surrounding materials as well as on the strengths of the materials in the interfacial transition zone. As shown in Figure 22, the fibers are tightly wrapped by the cement paste, which reflects the strong bonding between the steel fibers and the matrix, resulting in a strong interfacial bond and the formation of small cracks at the interface [43,44].

After carbonation, the plate-like calcium hydroxide and needle-rod ettringite particles disappear, yielding calcium carbonate, as shown in Figure 23. The size of the large hole in the concrete structure decreases, and the internal structure of the entire system is adjusted to increase the concrete density. The uncarbonated concrete surface is relatively rough, and the contact between the aggregate and the slurry easily produces cracks. In contrast, the surface of carbonated concrete is smooth; the interactions between the aggregates and the mud are strong, and the gap in the interfacial transition zone of the carbonized concrete is reduced by filling it with carbonation products. Therefore, carbonation can effectively inhibit the generation of concrete cracks, prevent corrosive substances from entering the concrete interior, strengthen the aggregate–slurry interactions, and improve the mechanical properties of concrete [45,46].

## 4. Conclusions

In this study, the mechanical and carbonation properties and microstructures of concrete specimens prepared at different iron tailings replacement rates, single-steel fiber-doped concrete, and hybrid steel–PVA fiber reinforced concrete were examined. From the obtained results, the following conclusions have been drawn:(1)After adding iron tailings to the concrete matrix, its compressive strength, splitting tensile strength, axial compressive strength, and elastic modulus increased. The compressive strength, splitting tensile strength, and axial compressive strength of the T30 iron tailings concrete were 7.6%, 17.7%, and 14.7% higher than those of ordinary concrete, respectively.(2)The addition of fibers further improved the mechanical and carbonation properties of the iron tailings concrete, and the effect of the steel–PVA fiber mixture was stronger than that of the single-doped steel fibers. With an increase in the PVA fiber content, the mechanical properties of the concrete deteriorated, and its carbonation depth decreased. Based on the obtained test results, a mathematical expression was derived for the stress–strain curve of the iron tailings concrete under compression, and the calculated values were in good agreement with the experimental data.(3)With an increase in the carbonation age, the compressive strength, splitting tensile strength, and axial compressive strength of the concrete increased. From the results of the carbonation test and a carbonation depth prediction model previously developed for ordinary concrete, a carbonation depth prediction model for the fiber-reinforced iron tailings concrete was established.(4)The replacement rate of iron tailings produced a significant impact on the concrete pore structure. When the replacement rate of iron tailings was 30%, the smallest intensity of the main peak in the T_2_ NMR spectrum was obtained, and the T2 spectral area decreased by 8.9%. The lowest total concrete porosity was achieved at this replacement rate, and its value was negatively correlated with the compressive strength.(5)With an increase in the carbonation age, the porosity of various concrete specimens decreased by 7.6%, 10.9%, 13.4%, and 15.1%, and the pore size decreased as well. Considering the existing pore size–compressive strength model, a pore size–compressive strength model was established by conducting a regression analysis of the NMR data.(6)The main purpose of this experiment was to explore the influence of the iron tailings replacement rate and the fiber content on the mechanical properties and carbonation properties of concrete. The test focused on the material properties, and the components are still unclear. In addition, the microstructure of this experiment was only tested by nuclear magnetic resonance and scanning electron microscopy. More advanced and more accurate instruments should be used to comprehensively explore the microstructure of hybrid fiber iron tailings concrete from different levels and angles, such as energy spectrum tests, CT scan tests, thermal analysis tests, etc.

## Figures and Tables

**Figure 1 materials-15-08992-f001:**
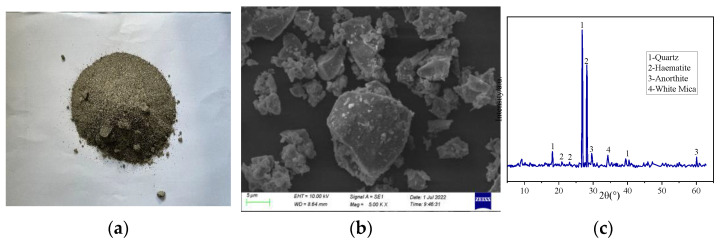
Microscopic analysis of iron tailings: (**a**) iron tail ore; (**b**) microscopic morphology of iron tailings; (**c**) iron tailings XRD.

**Figure 2 materials-15-08992-f002:**
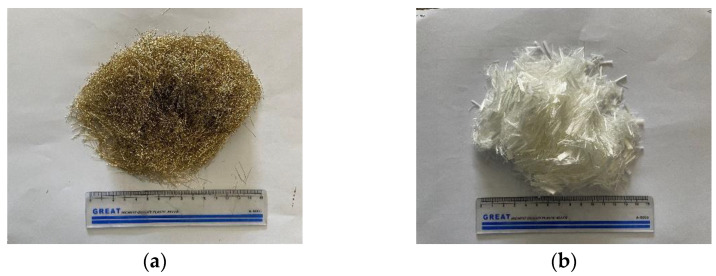
Fibers: (**a**) steel fiber; (**b**) PVA fiber.

**Figure 3 materials-15-08992-f003:**
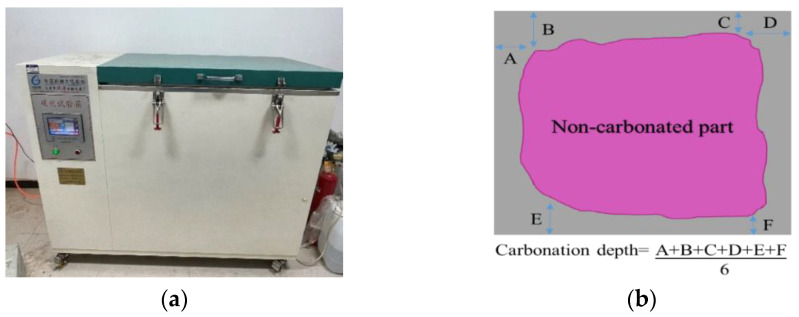
(**a**) Carbonization test chamber and (**b**) carbonation depth measurement.

**Figure 4 materials-15-08992-f004:**
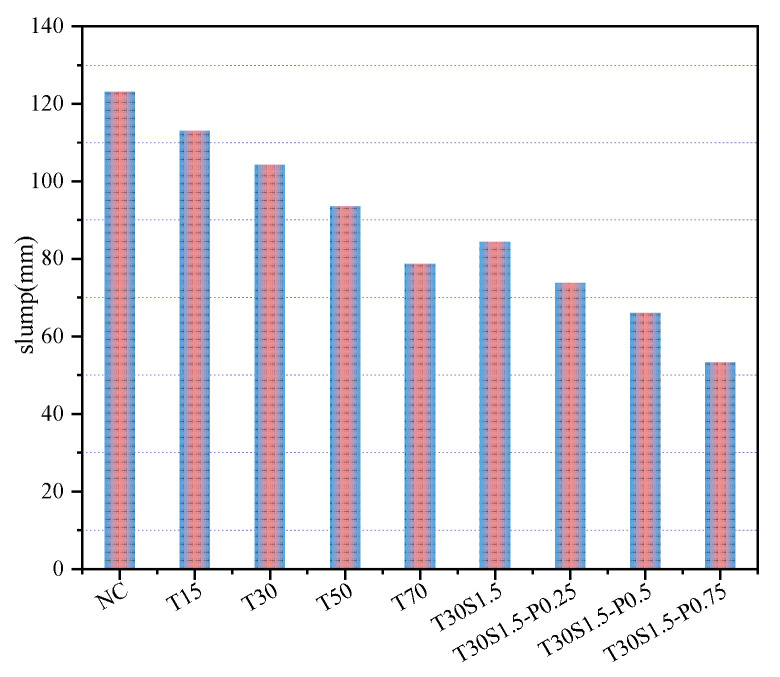
Concrete slump.

**Figure 5 materials-15-08992-f005:**
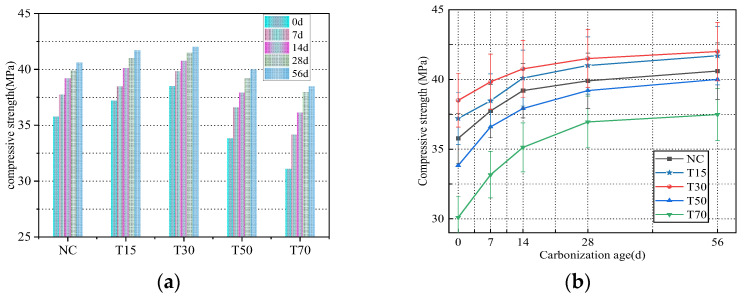
(**a**) Influence of replacement rate of iron tailings on compressive strength (**b**) Influence of carbonation age on compressive strength.

**Figure 6 materials-15-08992-f006:**
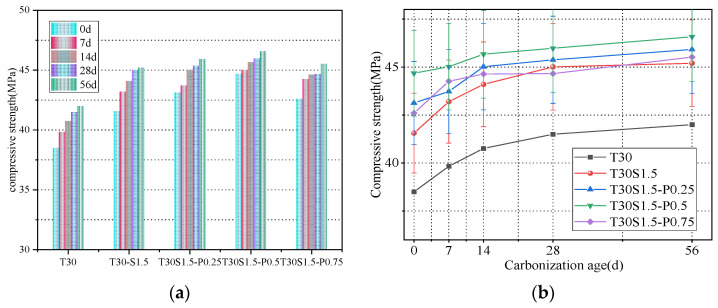
(**a**) Effect of fiber content on compressive strength (**b**) Influence of carbonation age on compressive strength.

**Figure 7 materials-15-08992-f007:**
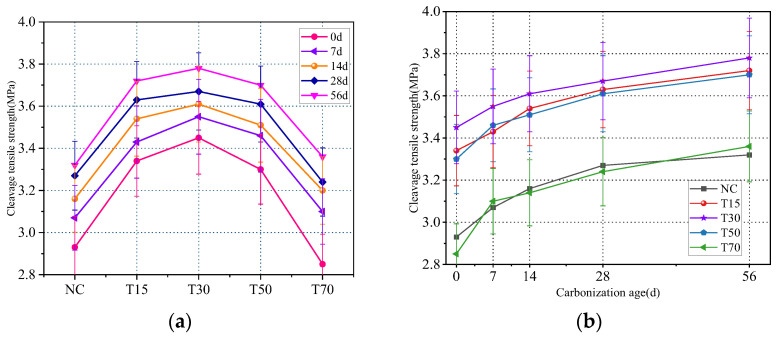
(**a**) Effect of replacement rate of iron tailings on splitting tensile strength; (**b**) Effect of carbonation age on splitting tensile strength.

**Figure 8 materials-15-08992-f008:**
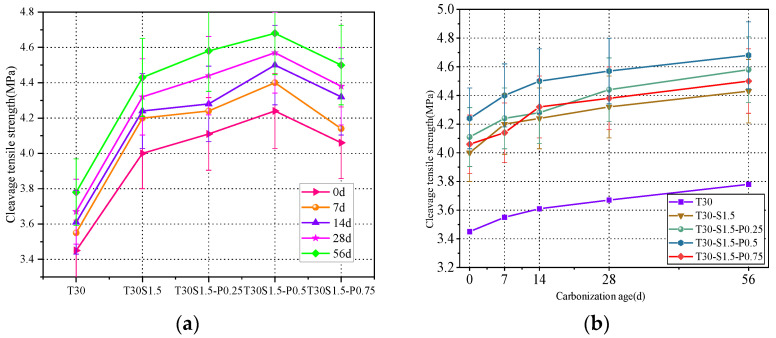
(**a**) Effect of fiber content on splitting tensile strength; (**b**) Effect of carbonation age on splitting tensile strength.

**Figure 9 materials-15-08992-f009:**
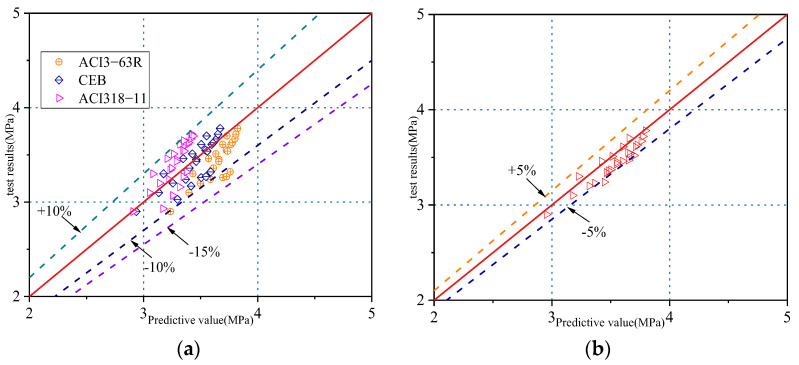
Relationship between tested and predicted values of splitting tensile strength. (**a**) Proposed constants in this test; (**b**) Using the constants in the standards.

**Figure 10 materials-15-08992-f010:**
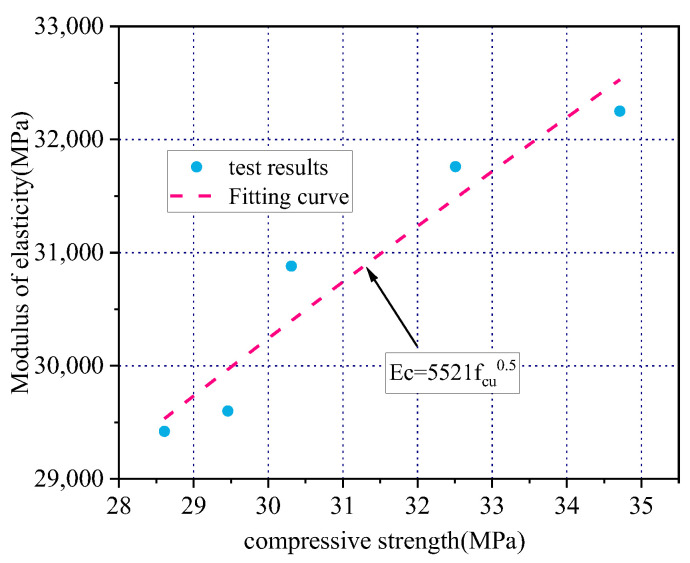
Proposed constants in this test.

**Figure 11 materials-15-08992-f011:**
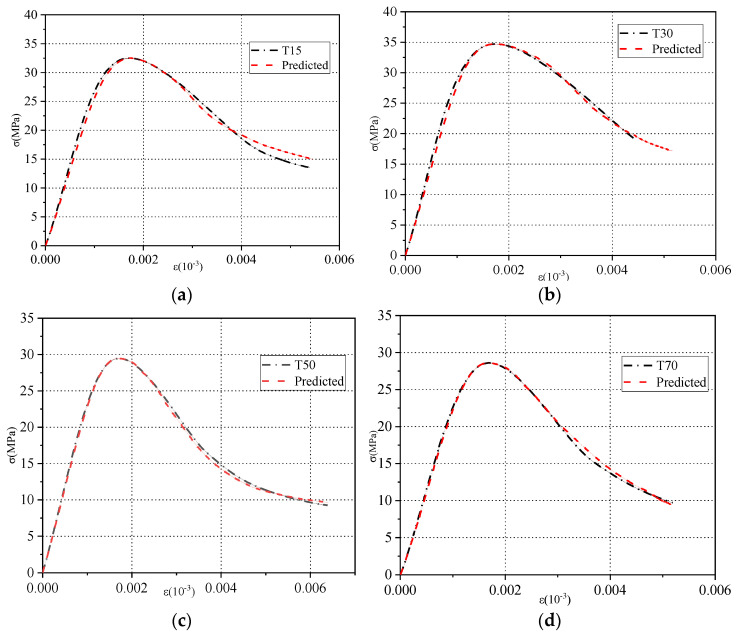
Stress−strain curve and prediction curve: (**a**) T15; (**b**) T30; (**c**) T50; (**d**) T70.

**Figure 12 materials-15-08992-f012:**
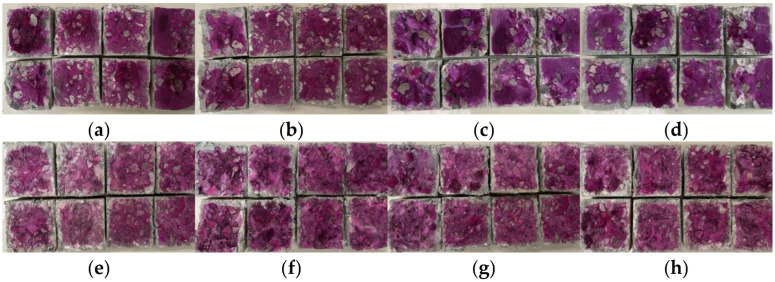
Concrete carbonation depth: (**a**) 15; (**b**) T30; (**c**) T50; (**d**) T70; (**e**) T30S1.5; (**f**) T30S1.5-P0.25; (**g**) T30S1.5-P0.5; (**h**) T30S1.5-P0.75.

**Figure 13 materials-15-08992-f013:**
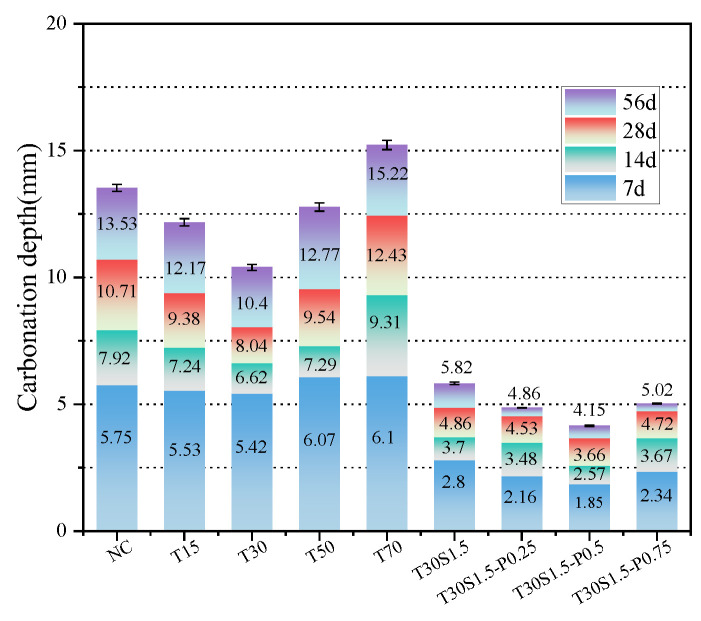
Effect of iron tailings substitution rate and fiber content on carbonation depth.

**Figure 14 materials-15-08992-f014:**
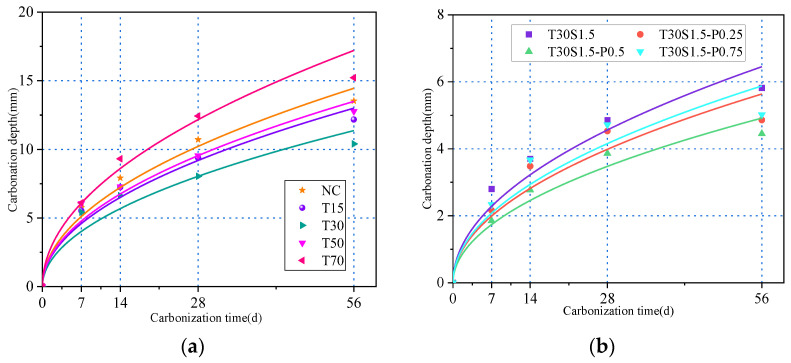
Fitting relationship between carbonation depth and time: (**a**) iron tailings; (**b**) fiber.

**Figure 15 materials-15-08992-f015:**
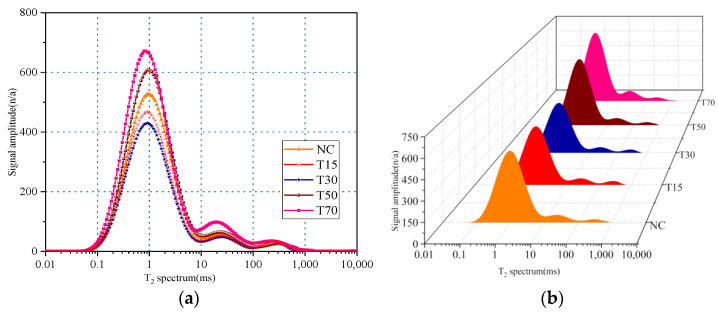
Effect of substitution rate of iron tailings on T_2_ spectrum of concrete: (**a**) two-dimensional spectra; (**b**) three dimensional spectra.

**Figure 16 materials-15-08992-f016:**
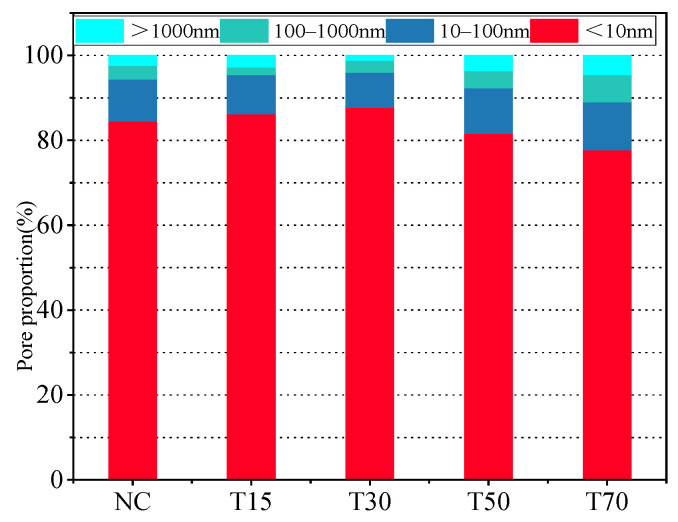
Pore size distribution of concrete with different replacement rates of iron tailings.

**Figure 17 materials-15-08992-f017:**
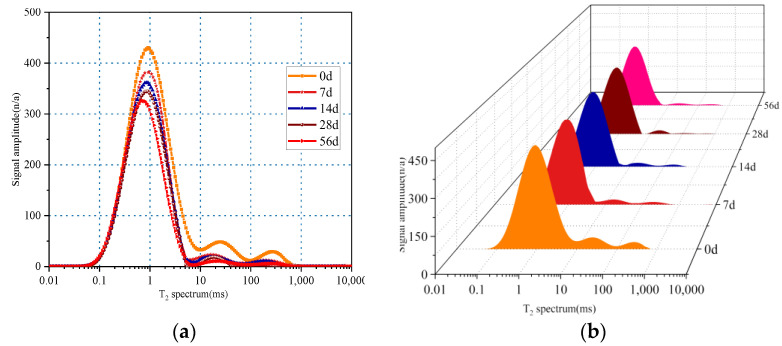
Effect of carbonation age on T_2_ spectrum of concrete: (**a**) two-dimensional spectra; (**b**) three-dimensional spectra.

**Figure 18 materials-15-08992-f018:**
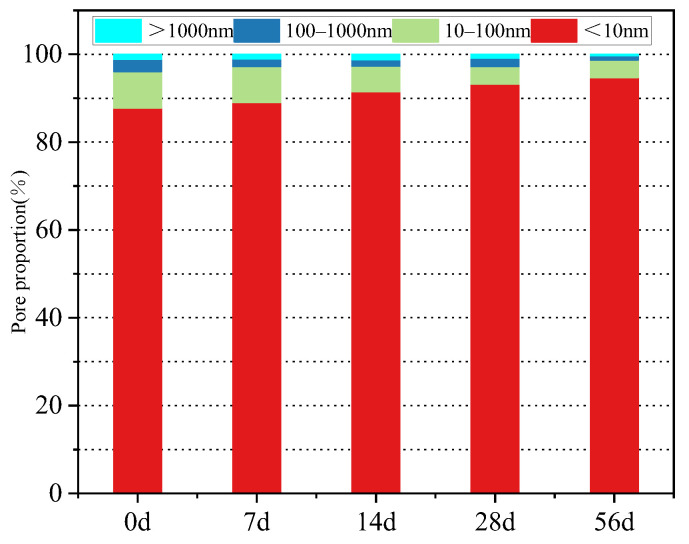
Pore size distribution of concrete at different carbonation ages.

**Figure 19 materials-15-08992-f019:**
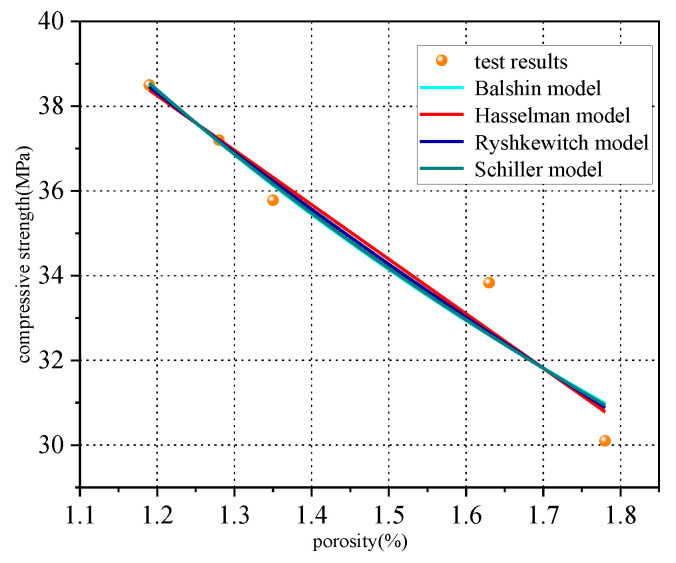
The relationship between porosity and compressive strength.

**Figure 20 materials-15-08992-f020:**
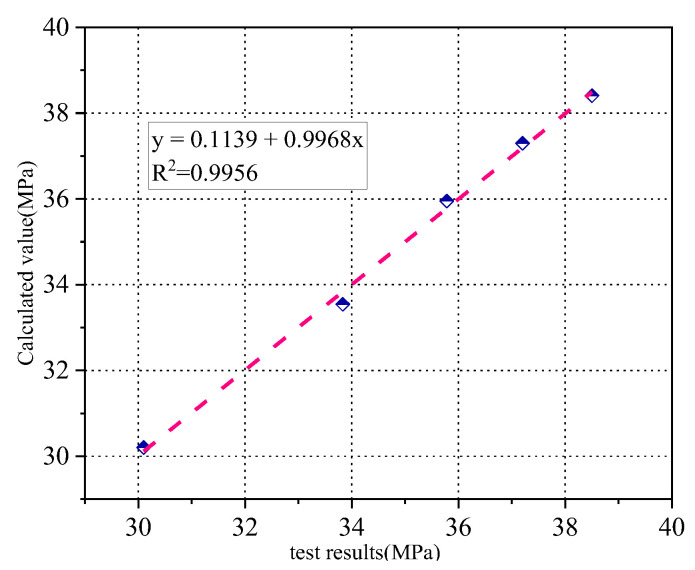
Comparison of compressive strength test values and calculated values.

**Figure 21 materials-15-08992-f021:**
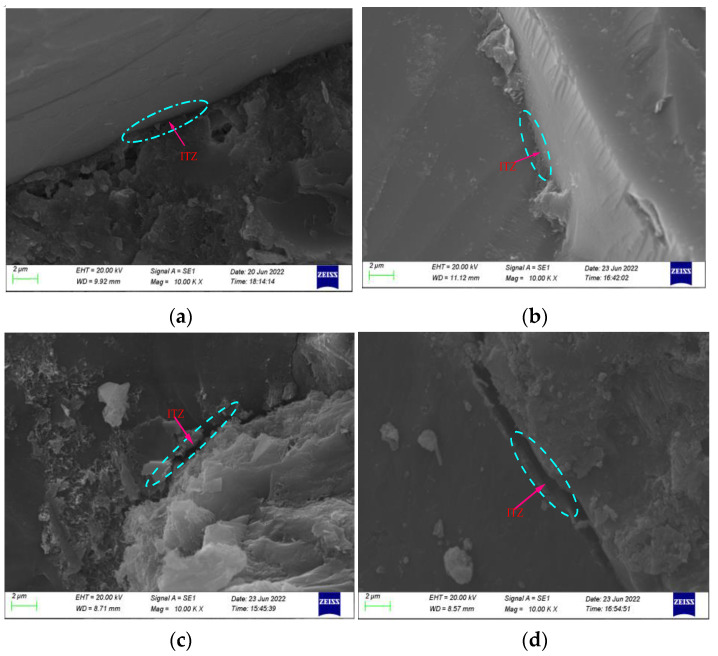
ITZ of iron tailings concretes: (**a**) NC; (**b**) T30; (**c**) T50; (**d**) T70.

**Figure 22 materials-15-08992-f022:**
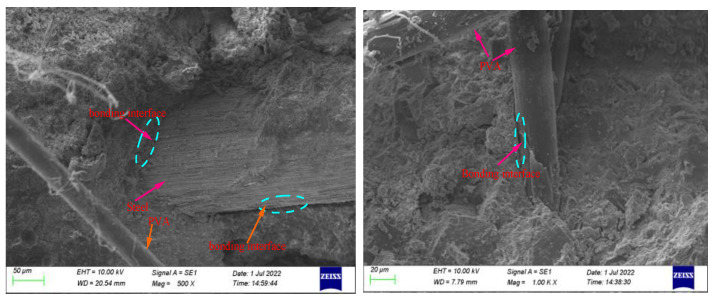
Fiber–slurry bonding interface.

**Figure 23 materials-15-08992-f023:**
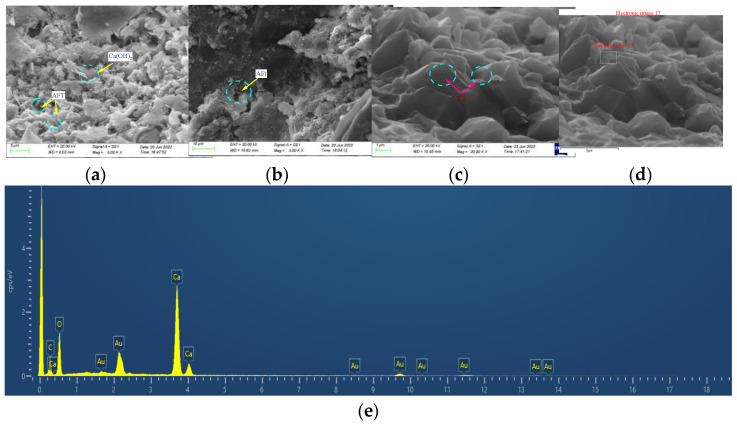
Micromorphology of calcium carbonate. (**a**) 0 d; (**b**) 7d (**c**) 28d; (**d**) CaCO_3_ (**e**) eds.

**Table 1 materials-15-08992-t001:** Chemical composition of cement and fly ash (mass fraction)/%.

	Al_2_O_3_	SiO_2_	MgO	CaO	TiO_2_	Fe_2_O_3_	SO_3_	Other
cement	42.5	55.7	/	0.4	0.9	0.3	/	0.2
fly ash	29.96	52.97	1.52	3.66	/	7.98	0.65	3.26

**Table 2 materials-15-08992-t002:** Physical properties of fly ash.

Moisture	Water Demand Ratio	Loss on Ignition	Density (g/cm^3^)	Fineness (45 μm Square Hole Sieve Allowance)
0.29	93	3.52	2.11	12

**Table 3 materials-15-08992-t003:** Physical properties of aggregates.

Aggregate Type	Loose Packing (kg/m^3^)	Apparent Area (kg/m^3^)	Water Absorption (%)	Void Ratio (%)	Crush Indicator (%)	Mud Content (%)
Gravel	1543	2769	0.63	40.10	10.30	0.59
River sand	1655	2689	0.5	/	/	3.1
Iron tailings sand	1824	2745	2.9	8.7	/	19.53

**Table 4 materials-15-08992-t004:** Physical properties of fibers.

Fiber Type	Diameter (mm)	Length (mm)	Density (g/cm^3^)	Aspect Ratio	Elastic Modulus (GPa)	Tensile Strength (MPa)
SF	0.22	16	7.8	72.73	210	2500
PVA	0.024	12	0.97–0.98	500	160	3000

**Table 5 materials-15-08992-t005:** Design of concrete mix ratio.

Block Coding	Fly Ash	Cement	Stone	River Sand	Iron Tailings Sand	SF	PV	Water	Water Reducer
NC	110	440	1103	656	0	0	0	220	2.2
T15	110	440	1103	557.6	98.4	0	0	220	2.2
T30	110	440	1103	459.2	196.8	0	0	220	2.2
T50	110	440	1103	328	328	0	0	220	2.2
T70	110	440	1103	459.2	196.8	0	0	220	2.2
T30-S1.5	110	440	1103	459.2	196.8	117	0	220	2.2
T30-S1.5P0.25	110	440	1103	459.2	196.8	117	2.45	220	2.2
T30-S1.5P0.5	110	440	1103	459.2	196.8	117	4.9	220	2.2
T30-S1.5P0.75	110	440	1103	459.2	196.8	117	7.35	220	2.2

Numerical naming: in “T30-S1.5-P0.25”, T represents iron tailings, S denotes steel fibers, P designates PVA fibers, “30” represents the iron tailings replacement rate of 30%, and “0.25” indicates the PVA fiber content of 0.25%.

**Table 6 materials-15-08992-t006:** Concrete *f*_cu_, *E*_c_, and *σ*_p._

Block Coding	Compressive Strength at Different Carbonization Ages					*E*_c_ (MPa)	*σ*_p_ (10^−3^)
	*f* _cu_	7d	14d	28d	56d		
NC	30.31	32.19	34.77	35.16	36.54	30,880	1.7
T15	32.51	34.46	36	37.15	37.65	31,760	1.72
T30	34.71	36.49	37.25	37.86	38.69	32,250	1.73
T50	29.46	32.86	34.04	34.25	35.67	29,600	1.68
T70	28.61	31.82	32.61	33.74	35.12	29,420	1.67
T30S1.5	37.4	39.31	40.52	42.55	43.49	33,460	2.21
T30S1.5-P0.25	38.53	39.86	42.06	43.18	44.54	35,220	2.54
T30S1.5-P0.5	39.82	41.77	42.83	43.86	45.29	36,050	2.93
T30S1.5-P0.75	37.94	40.56	41.32	42.03	43.97	34,830	2.61

**Table 7 materials-15-08992-t007:** Errors of different standard test values and calculated values.

	Equation (6)	ACI318-11	ACI363R
error rate	−0.37–1.57%	19.9–22.1%	15.8–18.7%

**Table 8 materials-15-08992-t008:** Fitting equation of carbonation depth and time.

Block Coding	Fitting Equation	β
NC	X = 1.93$\sqrt{t}$	25.57
T15	X = 1.74$\sqrt{t}$	23.04
T30	X = 1.52$\sqrt{t}$	20.14
T50	X = 1.80$\sqrt{t}$	23.85
T70	X = 2.33$\sqrt{t}$	30.48
T30S1.5	X = 0.86$\sqrt{t}$	12.62
T30S1.5-P0.25	X = 0.75$\sqrt{t}$	11
T30S1.5-P0.5	X = 0.69$\sqrt{t}$	10.12
T30S1.5-P0.75	X = 0.79$\sqrt{t}$	11.59

**Table 9 materials-15-08992-t009:** Ratios of calculated values to test values.

Block Coding	7d	14d	28d	56d
NC	0.84	0.84	0.86	0.94
T15	0.87	0.93	0.99	1.05
T30	0.89	1.02	1.17	1.25
T50	0.79	0.92	0.97	1.00
T70	0.98	0.98	1.01	0.96
T30S1.5	0.80	0.85	0.91	1.06
T30S1.5-P0.25	1.04	0.90	0.98	1.18
T30S1.5-P0.5	1.21	1.23	1.21	1.11
T30S1.5-P0.75	0.96	0.86	0.94	1.23

**Table 10 materials-15-08992-t010:** Area and proportion of NMR spectra before carbonization.

Block Coding	T2 Spectrum Area	Peak1		Peak2		Peak3	
		Area	Proportion	Area	Proportion	Area	Proportion
NC	6055.7	5273.9	87.1	593.4	9.8	187.7	3.1
T15	5670.5	5000.9	88.2	544.3	9.6	124.7	2.2
T30	5517.7	4943.2	89.6	502	9.1	71.7	1.3
T50	6960.6	6027.4	86.6	703	10.1	229.7	3.3
T70	8001.7	6856.9	85.7	856.1	10.7	288	3.6

**Table 11 materials-15-08992-t011:** NMR spectrum area and proportion.

Age	T2 Spectrum Area	Peak1		Peak2		Peak3	
		Area	Proportion	Area	Proportion	Area	Proportion
0	5517.7	4943.2	89.6	502	9.1	71.7	1.3
7	5266.1	4860.6	92.3	352.8	6.7	52.7	1
14	5088.6	4661.1	91.6	366.4	7.2	61.1	1.2
28	4970.3	4632.3	93.2	318.1	6.4	19.9	0.4
56	4911.1	4650.8	94.7	255.4	5.2	4.9	0.1

**Table 12 materials-15-08992-t012:** Models of the relationship between porosity and compressive strength.

Model [41]	Fitting Equation	R^2^
Balshin model	σ_c_ = 78.78(1−p) − 0.91	0.93516
Hasselman model	σ_c_ = 53.72 − 12.89p	0.94613
Ryshkewitch model	σ_c_ = 59.79exp(−0.37p)	0.94108
Schiller model	σ_c_ = 18.86ln (9.19/p)	0.9452

## Data Availability

The data presented in this study are available on request from the corresponding author.

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
