# Peer review of "Carbonation Resistance and Pore Structure of Mixed-Fiber-Reinforced Concrete Containing Fine Aggregates of Iron Ore Tailings"

_materials, 2022, doi:10.3390/ma15248992_

Round 1

Reviewer 1 Report

This manuscript evaluates the Carbonation resistance and pore structure of mixed fiber-reinforced concrete containing fine aggregates of iron ore tailings. The manuscript is elaborately described and contextualized with the help of previous and present theoretical background and empirical research. All the references cited are relevant to this area of research and also adequate. The methods are clearly stated. The result and discussion of the research are coherent and balanced. The conclusions are supported by the results.  However, some minor corrections need to be addressed before the acceptance the Manuscript.

1. What is the novelty of this research? Mention it.

2. Table 1. Mention the unit % for cement and fly ash

3. Provide clear image of figure 1. The magnification of SEM image is not readable.

4. Table 4. Mention the diameter in ‘mm’

5. How did you arrive this mix ratio?

6. Show the images of iron ore tailings, and the fibers used in this study.

7. The result of compressive strength need to be compared with existing literatures.

8. Which standards suggests Equation 1?

9. Line no.274. How the error 5% was obtained?

10. Fig 14. b. 16 b. axis title, axis values are not clearly visible

11. Fig.20. Provide clear images.

12. Mention your research recommendation at the end of conclusion part.

Author Response

Thank you very much for your valuable suggestions on this article. The following is my revision and reply to the proposal.

1. The novelty of this paper has been added to the introduction.

2. The percentage by mass of cement and fly ash has been listed in Table 1.

3. Figure 1. The image has been re provided.

4. The fiber diameter in Table 4 has been modified to "mm".

5. This paper refers to the requirements of Specification for Design of Ordinary Concrete Mixtures (JGJ55-2011), and the mixing ratio is obtained through pre test and adjustment. This is reiterated in document 2.2.

6. Iron tailings, steel fiber and PVA images are provided

7. The compressive strength results of the paper are compared with the existing literature.

8. Formula 1 is recommended by reference.

9. In line 274, formula 2 is obtained from linear regression analysis. Compare the calculated value of formula 2 with the experimental value, and the error is within ± 5%.

10. The titles of figures 14. b and 16. b have been readjusted.

11. As the current epidemic situation is too serious to carry out experiments, new images cannot be obtained. However, I have resized the image.

12. Research recommendations already mentioned at the end of the conclusion

The above are my changes to the proposal. Thank you again for your valuable suggestions on this article.

Reviewer 2 Report

1) “The results demonstrate that the compressive and splitting tensile strength of concrete first increase and subsequently decrease with the increase in the iron tailings replacement rate, while carbonation depth and porosity initially decrease and subsequently increase.” Please modify this abstract sentence. The abstract should contain the main results of the study. The sentence is vague and does not show, for example, the percentage increases in resistance and splitting tensile strength.

2) “and the relationship between concrete compressive strength and porosity was established.” This type of relationship has been widely reported in the literature. Therefore, this type of information does not need to be presented in the abstract of the work.

3) “As an engineering material commonly used in building structures, concrete exhibits high compressive strength but low tensile strength, ductility, and toughness, which are easily affected by the environment” Very general sentence. Modify by specifying that these characteristics are typically observed in conventional concrete.

4) In the introduction, it is necessary to include the main differences between steel and PVA fibers and highlight the main findings of existing studies with both materials. It is also necessary to clarify why the combined use of steel fibers and PVA can be interesting. This information is not presented in the manuscript.

5) “The fine aggregates were composed of Chanhe natural sand and iron tailings, and their main components were analyzed by scanning electron microscopy and X-ray diffraction.” Insert information about the analyzes mentioned, for example, the equipment used, the test conditions, among others.

6) Insert the code of the files and the database used in the diffractogram analysis shown in Figure 1.

7) Explain the characterization results presented in Table 3. For example, why does iron tailing sand have superior water absorption?

8) In section 2.2 the authors need to present more information about some dosing parameters used. As can be seen in the Table 5, the amount of water and superplasticizer is the same for all compositions. However, what is the impact of fiber incorporation and iron tailing sand on the fresh state of concrete (slump)? Why wasn't the slump fixed and the amount of superplasticizer adjusted? This type of information must be presented in the manuscript.

9) The fiber content was added based on what? In the cement weight? Make this clear in the materials and methods section.

10) What is the CO2 concentration adopted for the carbonation tests? Insert this information to the manuscript. Moreover, only six depth of carbonation readings were taken for each sample?

11) Remove Fig. 3. It does not bring important information to the study.

12) In Figures 4 and 5 insert the deviation bars of the compressive strength values. A statistical analysis of the results would also be interesting.

13) “As shown in Figure 4, when the replacement rate of iron tailings is equal to 30%, the 168 concrete compressive strength reaches a maximum, and its strength increases by 7.6%. 169 concerning that of ordinary concrete. The compressive strengths of the T15, T50, and 170 T70 concrete specimens increase by 4%, −5.8%, and −13.1%, respectively.” The observed variations are very small and, possibly, do not have significant statistical differences. Rephrase the sentence to show that few differences were observed.

14) “In addition, the water demand of the concrete mixture increases, and its fluidity decreases, leading to the separation of the mortar and aggregates, which is not conducive to compact forming; as a result, the strength of the iron tailing concrete decreases.” The slump values were not presented in the manuscript, although they are essential information for discussing the mechanical results and porosity.

15) “The compressive strength of concrete increases first and then decreases with an increase in the PVA fiber content, and the concrete strength increases by 7.9%, 12%, 16%, and 10.6%, respectively.” Check which of these differences are statistically significant.

16) “Steel fibers inhibit the formation of large cracks in concrete, and PVA fibers inhibit early plastic shrinkage, dry shrinkage, and microcrack generation.” Insert references that support this sentence.

17) Insert the standard deviations in Figure 6.

18) Based on the results, would it be possible to propose changes in the constants of the equations that correlate compressive strength and tensile strength?

19) “With an increase in the carbonation age, the secondary hydration reaction between iron tailings and Ca(OH)2, the main hydration product of cement, results in the formation of calcium silicate hydrate and other compounds, thus improving the internal pore structure and concrete density.” Does this information mean that iron tailing has pozzolanic activity? Is it possible to insert the material's chemical composition (XRF)? I believe it could help explain the interaction it can have with cement hydration products. Several studies in the literature did not confirm the pozzolanic activity of this residue. Review sentence. In addition, it is necessary to insert references that support the statement of the previously shown sentence. Furthermore, if iron tailing reacted with Ca(OH)2, it would reduce the alkaline reserve of the concretes and, therefore, the resistance to carbonation as a secondary effect to the porosity of the matrix.

20) “PVA fibers are hydrophilic, which promotes the accumulation of a large amount of water on their surface, creates favorable conditions for cement hydration, inhibits the early formation of plastic shrinkage and dry shrinkage cracks, reduces the cracking temperature, and increases the crack resistance of the concrete.” Insert references.

21) “After carbonation, the plate-like calcium hydroxide and needle-rod ettringite particles disappear, yielding calcium carbonate Figure 22.” Modify the discussion of the SEM images. To draw quantitative conclusions about the phases formed, it is necessary to analyze several images. It is impossible to reach this type of conclusion with few images.

22) How was CaCO3 identified in Figure 22? Through EDS?

Author Response

Thank you very much for your valuable suggestions on this paper. The following is my modification and reply to the suggestions.

1 The main findings of the study have been restated in the abstract.

2 ‘The relationship between compressive strength and porosity of concrete is established’, Deleted.

3 Concrete has been changed to ordinary concrete.

4 In the introduction, the main differences between steel and PVA fibers have been reinterpreted, and why steel fibers and PVA are combined has been explained.

5 Reinserted scanning electron microscope device information.

6 Code inserted into database in diffraction pattern analysis shown in Figure 2.

7 The characterization results given in Table 3 are the measurements given by the merchant when purchasing the material,as for why iron tailings have superior water absorption, it is because the water absorption rates of natural sand and iron tailings are 0.5 % and 2.9 %, respectively, as explained in the table.

8 The slump value has been added in the paper 3.1.

9 The steel fiber content is the optimal level when 1.5 % is obtained by referring to the existing research results of the project team and reading the references.

10 The concentration of carbon dioxide used in carbonation test is ( 20 ± 3 ) %, six carbonation depth readings were collected for each sample and averaged.

11 deleted.

12 The deviation bars of compressive strength values have been inserted in Fig.4 and Fig.5.

13 Re - comparison of the compressive strength results with references.

14 Slump value added in 3.1 of this article.

15 Re - comparison of the compressive strength results with references.

16 The literature citation has been reinserted.

17 The literature citation has been reinserted.

18 The change of equation constants related to compressive strength and tensile strength can be put forward by consulting literature and test results.

19 By reading the literature that iron tailings have volcanic ash activity, and insert the reference.

20 The literature citation has been reinserted.

21 A new image is inserted in Figure 22 to reflect the carbonation mechanism.

22 Caco 3 in Figure 22 was identified by EDS,a new image has been inserted.

The above are the modifications I made to the suggestions. Thank you again for your valuable suggestions on this paper.

Round 2

Reviewer 2 Report

1) Insert the code of the files and the database used (e.g., ICDD) in the diffractogram analysis shown in Figure 2.

2) Remove Fig. 1

Author Response

Thank you very much for your valuable suggestions on this paper. The following is my modification and reply to the suggestions.

1. Inserted

2. Removed

Above are my modifications to the suggestions. Thanks again for your valuable comments on this article.